# Short Tandem Repeat (STR) Profiling of Earwax DNA Obtained from Healthy Volunteers

Sayed Amin Amer *, Maha Nawar Alotaibi, Sajjad Shahid, Mahmoud Alsafrani and Abdul Rauf Chaudhary

Department of Forensic Sciences, College of Criminal Justice, Naif Arab University for Security Sciences, Riyadh 11587, Saudi Arabia; maha-n-@hotmail.com (M.N.A.); sshahid@nauss.edu.sa (S.S.); drarch1954@gmail.com (A.R.C.)
* Correspondence: samer@nauss.edu.sa; Tel.: +966-559822001

**Abstract:** The present study aimed to establish human earwax as a potential source of DNA evidence that could be effectively used in human identification. Sixty earwax samples were obtained from 15 healthy male and female Saudi volunteers living in Riyadh, Saudi Arabia. Four consecutive earwax swab samples were obtained from each volunteer and stored for 1, 15, 30 and 60 days. Earwax samples were stored at room temperature (20–22 °C). Reference oral swab was also taken from each volunteer. DNA was extracted by QIAamp DNA Mini kit and quantified by real-time polymerase chain reaction (RT-PCR) on 7500 Thermal Cycler. Autosomal STR loci were amplified using AmpFLSTR™ Identifiler™ Plus PCR Amplification Kit (Thermo Fisher Scientific, Carlsbad, CA, USA). Amplified fragments were size separated and analyzed on a 3500 Genetic Analyzer. Complete autosomal STR profiles were obtained from the earwax swabs of all the volunteers stored up to 30 days after the collection. Some STR profiles were partially obtained 60 days after the earwax collection. Allelic drop-out, allelic drop-in, and stutters were seen in earwax samples analyzed 60 days after the collection. The results have shown that human earwax can be a potential source of DNA evidence for human identification up to 30 days after the earwax collection. It is recommended to quickly analyze earwax samples or store them at room temperature or at −10 °C after their recovery from the crime scene.

**Keywords:** earwax DNA; cerumen; autosomal STRs; locus drop-out; human identification

## 1. Introduction

Earwax, also known as cerumen, is a viscous apocrine fatty secretion of ceruminous glands found in the skin epithelium of external ear canal. It usually contains keratinized cells and hairs [1,2]. Usually, the type of biological evidence used during forensic identification of human includes body fluids (blood, semen, saliva, vaginal fluid, urine, tears, and earwax), bone, teeth, tissues, hair, and a variety of trace-DNA specimens. Earphones might be used by offenders or suspects with mobile phones or MP3s attached to the outer ear, and thus, the outer ear earwax possibly sticks to these devices. Previous reports have shown that in the absence of commonly encountered biological traces at the crime scene, used earphones and other hearing devices containing traces of earwax can be effectively used in forensic casework [3–5] by employing mitochondrial DNA (mtDNA) sequencing of hypervariable regions 1 and 2 of human earwax mtDNA. Despite its forensic significance, published work regarding the use of human earwax for the purpose of autosomal STR genotyping is very limited [3–5].

Seo et al. [3], in addition to performing sequencing of mtDNA hypervariable regions 1 and 2 (HV-I and HV-II), also studied six autosomal STR markers from the earphones found at a crime scene. Although partial STR profiles were obtained from earphone earwax samples found at the crime scene, authors were able to obtain full DNA profiles in their experimental samples. A complete STR profile could be successfully typed when the amount of nuclear DNA recovered from volunteer earphones was ≥0.5 ng/μL [3]. The

presence of degraded DNA was attributed to the appearance of partial STR profiles. In another report, Yudianto et al. [4] performed sequencing of human earwax mtDNA HV-I and HV-II regions for the purpose of human identification in a criminal case. In their report, the authors studied human earwax sticking to the headphones used by the volunteers and found significant variations in the hypervariable regions of earwax mtDNA. Yudianto and Nzilibili [5] also collected DNA from the earwax stacked to the earphones and performed DNA sequencing of mtDNA for identification purposes.

Most of the studies conducted so far using human earwax as a possible source of DNA in forensic casework are based upon mtDNA sequencing of HV-I and HV-II regions, whereas reports employing autosomal STR genotyping of earwax sticked to earphones are very limited [3]. Studies using earwax swabs as a source of nuclear DNA face difficulties in producing complete STR profiles because of partial or complete DNA degradation caused by a variety of environmental factors such as moisture, temperature, bacterial growth, or other prevailing weather conditions, which may have a direct or indirect impact on the integrity of earwax DNA molecule.

Of the various artifacts produced by degraded earwax DNA typing, allelic drop-out, allelic drop-in, locus drop-out, and stutters are quite common [6,7]. Allele drop-out is an extreme example of heterozygote imbalance, where one allele falls below the limit of the detection threshold (LDT) [8]. "The stochastic threshold is the peak height above which it is reasonable to assume that allele drop-out of a sister allele of a heterozygote has not occurred at a locus" [9].

The presence of human earwax is possibly encountered in crime scenes in the form of used facial tissues, which are usually discarded into a dustbin or thrown on the ground after cleaning an itching ear. The present study aimed to explore the possibility of using human earwax swabs as a potential source of typable human DNA for the purpose of human identification in criminal cases. In the absence of other biological evidence, human earwax may provide a possible nuclear DNA source that could be used in forensic casework seeking human identification.

In addition, the effect of storage time on the nuclear DNA content is also examined. It is hoped that the outcome of this study will provide baseline data for future research on the use of human earwax in the forensic identification of individuals. The conventional autosomal STR genotyping protocol (extraction, quantification, and electrophoresis) were used in the present study because the DNA content of the crime scene earwax swab was not known.

## 2. Materials and Methods

Fifteen healthy Saudi volunteers consisting of 6 males and 9 females living in the city of Riyadh (KSA) were recruited for the present study. All the volunteers were explained the nature of this study and the number of samples required from each volunteer. After receiving complete information, volunteers provided their written informed consent to donate their earwax specimens. Four earwax specimens (day 1, 15, 30, and 60, respectively) were collected from each volunteer using a Puritan 6" sterile DNA-free standard cotton swab (Sterilin swab, Thermo Scientific, Carlsbad, CA, USA). Earwax samples were obtained either from the single ear or both the ears, depending upon the availability of the visible quantity of earwax. Each volunteer donated 4 earwax samples. In addition, a reference buccal swab was also obtained from each volunteer to verify the STR genotyping results. Earwax specimens were air dried at room temperature by removing the swab from the container and placing it outside at the ambient temperature (20–22 °C) for 2–3 h. After drying, samples were stored at room temperature until processed for DNA extraction and profiling. Day 1 specimens were processed after 24 h, while the remaining samples were processed 15, 30, and 60 days after their collection.

Before DNA extraction was carried out, the average weight of a blank cotton swab was determined so that the weight of earwax deposited on each cotton swab could be estimated, and approximately an equal quantity of earwax could be used in DNA extraction from all

swabs. Accordingly, earwax-containing portion of the swab was sliced from the parent wooden swab and used for DNA extraction. Blank cotton swabs were used as "control negative" with each DNA extraction, amplification, and electrophoresis.

The earwax-containing swab was cut into small pieces and placed in a 1.5 mL sterilized Eppendorf centrifuge tube. QIAamp® DNA Mini Kit (Qiagen, N.V., Venlo, The Netherlands) was used for the extraction of buccal swab and earwax nuclear DNA following the manufacturer's [10]. No modification was introduced in the procedure. The final elution was made in 200 μL of AE elution buffer. One reagent blank and one sterile DNA-free standard cotton swab were also extracted along with volunteers' swabs to cross-check any type of DNA contamination during the extraction and STR genotyping procedures.

The extracted DNA was quantified by a 7500 Real-Time PCR System (Applied Biosystems, Warrington, UK) using Quantifiler™ Duo DNA Quantification kit (Life Technologies, Foster City, CA, USA) according to the manufacturer's protocol. Amplification of 15 autosomal STR Loci (D8S1179, D21S11, D7S820, CSF1PO, D3S1358, TH01, D13S317, D16S539, D2S1338, D19S433, vWA, TPOX, D18S51, D5S818, and FGA) along with sex identification locus (Amelogenin) was performed in a Veriti Thermal Cycler (Thermo Fisher™, Waltham, Massachusetts, USA) using AmpFLSTR® Identifiler Plus® PCR Amplification kit [11]. Amplified fragments were size separated on Applied Biosystems 3500 Genetic Analyzer, and electrophoretic data was analyzed using GeneMapper™ ID-X Software v1.6 (Applied Biosystems, Foster City, CA, USA).

For DNA concentration, statistical analyses were performed using SPSS (version 28, IBM Corp., Armonk, NY, USA), and the significance was considered when *p*-value ≤ 0.05. According to Dytham [12] and Little [13], repeated measures ANOVA is used when the assumptions are met, and Friedman's test is used when the assumptions are not met. In the present study, we assumed that there was no normal distribution in the DNA concentrations extracted from the earwax samples, DNA concentrations at day 1, day 15, day 30, and day 60 was, therefore, tested by the Shapiro–Wilk at significant level $p < 0.05$. A Friedman test was used to detect the significant differences in mean DNA concentration rankings, and multiple comparisons were used along with a Bonferroni correction to find statistically significant differences between the data groups.

## 3. Results

The main objective of this study was to extract, quantify, and perform autosomal STR genotyping of the human earwax present in used cotton buds found at the crime scenes. For this study, we recruited 15 healthy male ($n = 6$) and female ($n = 9$) donors who provided us with their earwax swabs, which were stored at room temperatures (20–22 °C) for different time intervals ranging from 24 h to 60 days. Each volunteer donated four earwax swabs to be tested after 24 h (day one) and 15, 30, and 60 days, consecutively.

The mean ± SD (min-max) weight of earwax/swab sampled at different time intervals is shown in Table 1. The mean ± SD weight of earwax recovered from day 1 room-temperature-stored swabs was 8.358 ± 8.235 (0.5–22.04) mg. The mean ± SD for earwax swabs sampled after 15 and 30 days was 11.0 ± 8.2904 (2.87–30.52) mg, and 9.09 ± 8.5406 (1.0–29.5) mg, respectively. The median values are also provided in the table. The maximum amount of earwax was recovered from swabs sampled after a time interval of 60 days which showed a mean ± SD earwax quantity of 20.84 ± 15.0748 (1.59–55.78) mg (Table 1).

Table 2 shows the mean ± SD (min-max) concentration of genomic DNA extracted from the volunteer's reference buccal swabs as well as volunteers' earwax swabs collected and stored from day 1 to day 60. The mean DNA concentration in 15 volunteers' reference buccal swabs was 2.4163 ± 2.6908 (0.10446–9.074) ng/μL. On the other hand, the mean ± SD (min-max) DNA concentration in volunteer's earwax gradually decreased from 0.1040 ± 0.1396 (0.010010–0.48871) ng/μL to 0.0009 ± 0.0023 (0.000–0.0091) ng/μL, analyzed after 1, 30, and 60 days of storage at room temperature, respectively.

**Table 1.** Mean ± SD weight (mg) of earwax/swab stored at room temperature for various durations.

| Statistical Parameter | Weight of Earwax | | | |
|---|---|---|---|---|
| | Day 1 | Day 15 | Day 30 | Day 60 |
| Mean | 8.4 | 11.0 | 9.1 | 20.8 |
| Median | 3.1 | 9.5 | 7.4 | 19.1 |
| SEM | 2.1 | 2.1 | 2.2 | 4.1 |
| SD | 8.2 | 8.3 | 8.5 | 15.1 |
| Minimum | 0.5 | 2.9 | 1.0 | 1.6 |
| Maximum | 22.0 | 30.5 | 29.5 | 55.8 |

**Table 2.** Mean ± SD concentration (ng/μL) of DNA extracted from earwax swabs stored for various durations.

| Statistical Parameter | Reference Sample | Earwax Samples | | | |
|---|---|---|---|---|---|
| | | Day 1 | Day 15 | Day 30 | Day 60 |
| Mean | 2.4163 | 0.1040 | 0.0290 | 0.0099 | 0.0009 |
| Median | 1.0931 | 0.0482 | 0.02125 | 0.0052 | 0.00003 |
| SEM | 0.6725 | 0.0361 | 0.0076 | 0.0028 | 0.0006 |
| SD | 2.6903 | 0.1396 | 0.0294 | 0.0109 | 0.0023 |
| Minimum | 0.1406 | 0.0010 | 0.0025 | 0.0004 | 0.00 |
| Maximum | 9.074 | 0.4871 | 0.104 | 0.0417 | 0.0091 |

According to the Shapiro–Wilk test ($p < 0.05$), DNA concentrations at day 1, 15, 30, and 60 were not normally distributed. The DNA concentration was significantly different at the different time intervals ($\chi^2$ (3) = 36.36, $p < 0.001$, Table 3). Post hoc pairwise comparisons were performed by a Friedman test with a Bonferroni correction for multiple comparisons (Table 4) and revealed the adjusted significant difference in the DNA concentration between day 1 and day 30 ($p < 0.018$), day 1 and day 60 ($p < 0.000$), day 15 and day 60 ($p < 0.001$), and day 30 and day 60 ($p < 0.018$).

**Table 3.** Friedman test for comparing DNA concentration across sixty days of storage at room temperature.

| Time | Mean Rank | N | Chi-Square | df | Asymptotic Significance |
|---|---|---|---|---|---|
| R_Con_1day | 3.80 | | | | |
| R_Con_15day | 2.80 | | | | |
| R_Con_30day | 2.40 | 15 | 36.36 | 3 | 0.0001 |
| R_Con_60day | 1.00 | | | | |

Table 5 represents the autosomal STR profile of reference buccal swabs (from 15 Saudi volunteers) stored at room temperature for 1 day and the autosomal STR profiles of earwax swabs stored at room temperature for 1 day up to 30 days after sample collection.

**Table 4.** Pairwise comparisons with a Bonferroni correction for multiple tests. Each row tests the null hypothesis that the Sample 1 and Sample 2 distributions are the same. Asymptotic significances (2-sided tests) are displayed. The significance level is 0.05. Significance values have been adjusted by the Bonferroni correction for multiple tests.

| Sample 1 | Sample 2 | Test Statistic | Std. Test Statistic | Sig. | Adjusted Sig. |
|---|---|---|---|---|---|
| R_Con_15day | R_Con_1day | 1.000 | 2.121 | 0.034 | 0.203 |
| R_Con_30day | R_Con_1day | 1.400 | 2.970 | 0.003 | 0.018 |
| R_Con_60day | R_Con_1day | 2.800 | 5.940 | 0.001 | 0.000 |
| R_Con_30day | R_Con_15day | 0.400 | 0.849 | 0.396 | 1.000 |
| R_Con_60day | R_Con_15day | 1.800 | 3.818 | 0.001 | 0.001 |
| R_Con_60day | R_Con_30day | 1.400 | 2.970 | 0.003 | 0.018 |

Complete STR profiles were obtained in all the buccal swab samples at the 15 STR loci and the gender locus Amelogenin except for two samples (No. 4 and 6) which showed null alleles at the STR locus D18S51. After re-scanning the demographic data, it was found that those two samples belonged to two real brothers who were not twins. The autosomal STR profiles of earwax swabs stored at room temperature for 1 day up to 30 days after sample collection are assembled in Table 5, as all samples furnished a complete autosomal STR profile at 16 loci including the gender locus, Amelogenin. Like buccal swabs, null alleles appeared at the same locus D18S51 in samples no. 4 and 6. Looking at the complete STR profiles of 15 earwax samples, no sign of DNA degradation such as allelic drop-out was seen in earwax samples stored for 15 and 30 days at room temperature.

STR profiles of earwax samples after 60 days of storage at room temperature are shown in Table 6. Of the total 15 samples, 7 (46.67%) did not produce a profile, either complete or partial, indicating absolute degradation of 15 autosomal STR loci, except in sample 1, where Amelogenin was detected. Two out of fifteen samples (13.33%) revealed a complete DNA profile showing microsatellite stability at all the STR loci in these samples. One tri-allelic pattern (12, 13, 14) appeared in sample 4 at the locus D8S1179, whereas the reference buccal swab of this volunteer showed the genotype 12, 13 at the same locus. Complete loss of either of the two alleles (allelic loss) was observed in 21 alleles at different STR loci (Table 6) in six samples. Allelic alterations were seen in four samples—in sample 8, where homozygous allele 15, 15 changed to heterozygous alleles 11, 15 at the locus D19S433; and in sample 9, where homozygous alleles 13, 13 converted to heterozygous alleles 7, 13 at the locus D5S818. Similar allele conversions were also seen in samples 9 and 10 at the locus D7S820 and D13S317, where homozygous allele 10, 10 changed to 10, 14 and heterozygous allele 12, 13 converted to 12, 10, respectively.

**Table 5.** Autosomal STR profile of reference samples (from 15 Saudi volunteers) stored for 1 day at room temperature. STR profiles of earwax samples (from 15 Saudi volunteers) after room temperature storage for 1, 15, and 30 days are the same.

| STR loci | Samples | | | | | | | | | | | | | | |
|---|---|---|---|---|---|---|---|---|---|---|---|---|---|---|---|
| | 1 | 2 | 3 | 4 | 5 | 6 | 7 | 8 | 9 | 10 | 11 | 12 | 13 | 14 | 15 |
| D8S1179 | 15, 15 | 13, 15 | 15, 15 | 12, 13 | 12, 13 | 12, 13 | 14, 15 | 15, 15 | 14, 14 | 15, 15 | 13, 14 | 14, 15 | 15, 15 | 14, 14 | 15, 15 |
| D21S11 | 28, 28 | 31, 31.2 | 31.2, 32.2 | 28, 32.2 | 28, 32.2 | 28, 31 | 28, 32.2 | 31, 32.2 | 31.2, 32.2 | 31, 31.2 | 30, 31.2 | 28, 28 | 31.2, 32.2 | 30.2, 32.2 | 31.2, 32.2 |
| D7S820 | 10, 12 | 9, 11 | 8, 11 | 9, 9 | 9, 9 | 8, 9 | 10, 10 | 8, 8 | 10, 10 | 8, 8 | 10, 11 | 10, 10 | 8, 10 | 10, 10 | 8, 13 |
| CSF1PO | 11, 11 | 11, 12 | 12, 12 | 10, 11 | 12, 13 | 11, 12 | 11, 11 | 11, 12 | 11, 12 | 12, 12 | 11, 12 | 11, 11 | 11, 11 | 9, 11 | 12, 12 |
| D3S1358 | 15, 17 | 17, 18 | 16, 17 | 17, 17 | 15, 17 | 17, 17 | 17, 18 | 17, 18 | 17, 18 | 17, 17 | 17, 17 | 17, 18 | 17, 18 | 15, 17 | 14, 16 |
| TH01 | 6, 6 | 6, 6 | 6, 10 | 6, 9 | 6, 10 | 9, 10 | 6, 6 | 6, 6 | 6, 6 | 6, 7 | 6, 10 | 6, 6 | 6, 10 | 6, 9 | 9.3, 10 |
| D13S317 | 11, 12 | 8, 13 | 11, 12 | 11, 12 | 11, 12 | 11, 12 | 11, 11 | 11, 12 | 11, 12 | 12, 13 | 8, 11 | 11, 12 | 11, 12 | 8, 11 | 11, 12 |
| D16S539 | 9, 13 | 11, 12 | 9, 12 | 8, 9 | 9, 11 | 8, 9 | 9, 12 | 9, 11 | 9, 12 | 11, 12 | 12, 13 | 12, 12 | 9, 12 | 12, 12 | 9, 12 |
| D2S1338 | 17, 20 | 17, 20 | 17, 20 | 20, 24 | 20, 24 | 19, 24 | 17, 20 | 20, 20 | 17, 24 | 20, 20 | 17, 20 | 17, 24 | 17, 24 | 17, 20 | 20, 20 |
| D19S433 | 16, 16.2 | 12, 15 | 14, 15 | 13.2, 14 | 14, 15.2 | 13.2, 15.2 | 15, 16 | 15, 15 | 13, 13 | 14, 15 | 12, 15 | 15, 16 | 13, 14 | 16, 16 | 15, 15 |
| VWA | 16, 18 | 18, 18 | 16, 18 | 18, 19 | 15, 19 | 15, 19 | 15, 16 | 15, 19 | 15, 19 | 16, 19 | 16, 18 | 15, 16 | 15, 19 | 16, 16 | 18, 19 |
| TPOX | 8, 9 | 8, 9 | 8, 9 | 8, 9 | 8, 9 | 8, 8 | 8, 9 | 8, 8 | 8, 9 | 8, 9 | 9, 9 | 8, 8 | 8, 8 | 8, 11 | 8, 9 |
| D18S51 | 13, 14 | 15, 15 | 13, 15 | Null | 12, 12 | Null | 12, 13 | 15, 15 | 12, 13 | 15, 16 | 15, 16 | 13, 14 | 15, 15 | 12, 14 | 13, 15 |
| Amelogenin | X, Y | X, Y | X, Y | X, Y | X, Y | X, Y | X, X | X, X | X, X | X, X | X, X | X, X | X, X | X, X | X, X |
| D5S818 | 10, 11 | 11, 13 | 10, 13 | 11, 11 | 11, 13 | 11, 13 | 9, 10 | 9, 10 | 13, 13 | 9, 13 | 11, 13 | 10, 11 | 9, 13 | 9, 9 | 12, 13 |
| FGA | 23, 24 | 23, 24 | 20, 23 | 23, 26 | 22, 24 | 23, 26 | 23, 24 | 23, 24 | 23, 24 | 24, 24 | 20, 24 | 23, 24 | 20, 23 | 23, 24 | 20, 23 |

**Table 6.** Autosomal STR profile of earwax samples (from 15 Saudi volunteers) after room temperature storage for 60 days.

| STR loci | Samples | | | | | | | | | | | | | | |
|---|---|---|---|---|---|---|---|---|---|---|---|---|---|---|---|
| | 1 | 2 | 3 | 4 | 5 | 6 | 7 | 8 | 9 | 10 | 11 | 12 | 13 | 14 | 15 |
| D8S1179 | - | 13, 15 | 15, 15 | 12, 13, 14 | - | 0, 13 | - | 15, 15 | 14, 14 | 15, 15 | - | - | 14, 0 | - | - |
| D21S11 | - | 31, 31.2 | 31.2, 32.2 | 28, 32.2 | - | 0, 31 | - | 31, 0 | 31.2, 32.2 | 31, 31.2 | - | - | 31.2, 0 | - | - |
| D7S820 | - | 9, 11 | 8, 11 | 9, 9 | - | 0, 9 | - | 8, 8 | 10, 14 | 8, 8 | - | - | 8, 10 | - | - |
| CSF1PO | - | 11, 12 | 12, 12 | 10, 11 | - | 0, 12 | - | 11, 12 | 11, 12 | 12, 12 | - | - | 11, 11 | - | - |
| D3S1358 | - | 17, 18 | 16, 17 | 17, 17 | - | 17, 17 | - | 17, 18 | 17, 18 | 17, 17 | - | - | 17, 18 | - | - |
| TH01 | - | 6, 6 | 6, 10 | 6, 9 | - | 9, 10 | - | 6, 6 | 6, 6 | 0, 7 | - | - | 6, 10 | - | - |
| D13S317 | - | 8, 13 | 11, 12 | 11, 12 | - | 11, 12 | - | 0, 12 | 11, 12 | 12, 10 | - | - | 11, 12 | - | - |
| D16S539 | - | 11, 12 | 9, 12 | 8, 9 | - | 0, 9 | - | 9, 0 | 0, 12 | 11, 12 | - | - | 9, 12 | - | - |
| D2S1338 | - | 17, 20 | 17, 20 | 20, 24 | - | 19, 24 | - | 20, 20 | 17, 24 | 20, 20 | - | - | 17, 24 | - | - |
| D19S433 | - | 12, 15 | 14, 15 | 13.2, 14 | - | 13.2, 0 | - | 11, 15 | 13, 13 | 14, 15 | - | - | 13, 14 | - | - |
| VWA | - | 18, 18 | 16, 18 | 18, 19 | - | 15, 0 | - | 15, 0 | 15, 0 | 16, 19 | - | - | 15, 19 | - | - |
| TPOX | - | 8, 9 | 8, 9 | 8, 0 | - | 8, 8 | - | 8, 8 | 8, 0 | 8, 9 | - | - | 8, 8 | - | - |
| D18S51 | - | 15, 15 | 13, 15 | Null | - | Null | - | 15, 15 | - | 15, 16 | - | - | 15, 15 | - | - |
| Amelogenin | X, Y | X, Y | X, Y | X, Y | - | X, Y | - | X, X | X, X | X, X | - | - | X, X | - | - |
| D5S818 | - | 11, 13 | 10, 13 | 11, 11 | - | 0, 13 | - | 9, 10 | 7, 13 | 9, 13 | - | - | 0, 13 | - | - |
| FGA | - | 23, 24 | 20, 23 | 0, 26 | - | 23, 26 | - | 23, 24 | 23, 24 | 24, 24 | - | - | - | - | - |

Allelic loss █ , Tri-allelic pattern █ , Allelic alteration █ .

## 4. Discussion

Earwax is one of those biological samples that contains small amounts of intact DNA. When an earwax swab is found at the crime scene, a variety of environmental factors might have played their role in affecting its efficiency as biological evidence. This necessitates absolute care in handling earwax samples so that a complete or conclusive STR profile could be obtained. To the best of our knowledge, no data has been published so far on typing DNA from earwax that could identifying the suspects. We therefore consider this body fluid among the biological traces facing gaps in knowledge on DNA transfer, persistence, prevalence, and recovery [14,15]. Our literature survey showed that very few studies were conducted to evaluate human earwax as a probable source of nuclear or mitochondrial DNA in forensic casework [3–5]. The sequencing of mtDNA depended on its hypervariable regions 1 and 2, which are used in human identification. The only study to obtain six autosomal STR loci from human earwax DNA was that of Seo et al. [3].

Repeated measures analysis corresponding to time intervals of 1, 15, 30, and 60 days has been used. The appropriate statistical method for the obtained data is either repeated measures ANOVA, when the DNA concentrations are normally distributed, or Friedman's test, when the concentrations are not normally distributed [12,13]. For the current data, Shapiro–Wilk test was significant, indicating that the data does not follow a normal distribution. We therefore conducted a Friedman test, which indicated that significant differences in DNA concentration over the period of the study (day 1, 15, 30, and 60, respectively) were found.

While handling low quantity or degraded earwax DNA samples, appropriate DNA extraction, quantification, and amplification kits should be used to obtain the best possible STR genotyping results. Using the QIAamp DNA Mini extraction kit and AmpFLSTR™ Identifiler™ Plus PCR Amplification Kit, a complete STR profile was obtained from 0.3 ng/µL earwax DNA up to 30 days after the collection and room temperature storage. Seo et al. [3] were able to obtain a complete autosomal STR profile from 0.5 ng/µL DNA extracted by phenol-chloroform from freshly collected earphone samples and amplified with two triplex systems (CTT and FFv Multiplexes, Promega, Madison, WI, USA).

Yudianto and Nzilibili [5] used Mag-Bind® Blood & Tissue DNA HDQ 96 DNA extraction Kit (Omega Bio-tek, Norcross, GA, USA) and were able to obtain 0.16 ng/µL DNA after 20 days of earwax collection. In the present study, we used a very common DNA extraction kit and were able to obtain 0.0099 ng/µL of DNA along with a complete STR profile after room temperature (20–22 °C) storage for 30 days. Sixty days of storage of earwax swabs at room temperature caused considerable DNA degradation resulting in the loss of several loci, except in samples 1 and 2, where complete STR profiles were obtained. This shows that samples stored at room temperature for more than 30 days could be treated as low-copy-number (LCN) samples. However, we were not able to do that in the present study because of certain limitations. The effect of prolonged storage on the quality of earwax DNA is also witnessed as the DNA concentration gradually decreased. Under such situations, extraction, quantification, amplification, and detection protocols for LCN DNA should be followed.

STR artifacts such as allele drop-out, stutter product formation, allele drop-in, allele slippage, and other microvariations are outcomes of various types of genetic variations taking place at the molecular level in vivo [6,7]. Some of the STR profiles of earwax samples, after 60 days of storage at room temperature, showed locus drop-out, a single allele drop-out or drop-in, or a stutter in others.

The risk of allelic drop-out increases for samples with low amounts of DNA. This phenomenon may be manifested as few input cells, causing the entire DNA profile to be weak, or degradation of the biological material, which may cause some STR molecules (often the longer sequences) to be present in a low quantity [16]. Hence, the post-PCR result may be a DNA profile with some alleles missing; e.g., some alleles fail to emit sufficient fluorescent signal for the associated peak intensity to be above a detection threshold (50 RFU). As the number of molecules reduces by about 100–200 pg [17], stochastic variation

could be possible, which is first evidenced by a change in peak height balance, because the peak height is no longer representative of the number of alleles in the sample. Eventually, the allele is not amplified and is manifested as an allelic drop-out. The peak heights of the alleles recorded in samples 4, 6, 8, 9, 10, and 13 after 60 days and highlighted in orange (Table 6) fall below the stochastic threshold (it was set properly to rfu 175), and therefore, drop-out events have possibly occurred. The homozygous genotypes below this ST were identified as likely allelic drop-outs. Stuttering is a different phenomenon. Different labs encounter different levels of a stutter, so we may infer that there is something in the process that either encourages or discourages stutter. It would be expected that a 'slippage' occurred early in the amplification due to a transcription error; however, as noted in [18], no experimental evidence so far supports this expectation. Drop-in is simply the introduction of extraneous DNA. The unusual aspect is that it appears to involve only a few loci in any sample. The reason for this is unclear, but recent work on extra-cellular DNA may shed some light on the source of these fragments [19]. Generally, the differentiation of drop-in and drop-out events has never been satisfactorily resolved due to the stochastic anomalies linked to STR typing of low-template DNA samples. The development of a logistic regression model to quantify the probabilities associated with the occurrences of these events was proposed by Gill et al. [8] and evaluated with alternative models [20].

The appearance of null alleles is not a strange phenomenon in STR-based forensic DNA profiling. This mutation can cause a complete lack of production of the associated gene product or a product that does not function properly. A null allele cannot be distinguished from deletion of the entire locus solely from phenotypic observation. In the present study, the autosomal locus D18S51 disappeared entirely in two brothers' (locus drop-out) earwax swabs stored at room temperature from day 1 to day 60. This phenomenon seems not to relate to sampling type or storage condition as it was also exhibited in their reference oral swabs. Mutation could be possible within the D18S51 sequence as this locus is characterized by a high mutation rate and wide allelic range [20,21], having more allelic drop-outs. Mutation may also occur in the flanking region where the primers anneal, thereby inhibiting locus amplification [22]. Degenerated primers may possibly amplify D18S51 [23].

## 5. Conclusions

Complete STR profiles were obtained from healthy volunteers' earwax swabs containing approximately 25–30 pg uncontaminated DNA after up to 30 days of storage at room temperature. On the other hand, we could not obtain a complete or partial autosomal STR profile, including the Amelogenin marker, in about 47% of our studied samples after storage of 60 days at room temperature. We therefore suggest using the best possible DNA quantification, amplification, and detection protocols for DNA profiling if the extracted earwax DNA concentration is less than 25 pg. Moreover, crime scene earwax swabs should be dried immediately and stored at room temperature (20–22 °C) or at −10 °C until DNA profiling is performed. Due to certain limitations, we were unable to study the effect of environmental factors (temperature, humidity, UV, and sunlight) on the behavior of earwax DNA recovered from outdoor crime scenes. We therefore strongly recommend conducting further studies on the evaluation of environmentally degraded earwax DNA employing an LCN protocol.

**Author Contributions:** Conceptualization, S.A.A. and A.R.C.; methodology, M.N.A. and S.S.; formal analysis, M.N.A. and M.A.; writing—original draft preparation, S.A.A. and A.R.C.; writing—review and editing, S.A.A. and A.R.C.; supervision, S.A.A. All authors have read and agreed to the published version of the manuscript.

**Funding:** This research received no external funding.

**Institutional Review Board Statement:** The study was conducted in accordance with the Declaration of Helsinki, and approved by the Institutional Review Board of Naif Arab University for Security Sciences (NAUSS) via the notification No: Nauss-Rec-22-04.

**Informed Consent Statement:** Informed consent was obtained from all subjects involved in the study. Written informed consent has been obtained from the participants to publish this paper.

**Data Availability Statement:** Data supporting the reported results can be obtained from M Alotaibi upon request.

**Acknowledgments:** We acknowledge Naif Arab University for Security Sciences for providing the facilities to conduct this research.

**Conflicts of Interest:** The authors declare no conflict of interest.

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
