# Peer review of "Short Tandem Repeat (STR) Profiling of Earwax DNA Obtained from Healthy Volunteers"

_cimb, doi:10.3390/cimb45070362_

Round 1

Reviewer 1 Report

The authors describe human earwax as potential biological source for forensic individual identification. In addition, they tested the stability of the DNA recovered from this specimen over time, up to 60 days from the collection. Full profiles matching the reference samples could be achieved up to 30 days after collection and storage at room temperature while locus dropouts and allelic drop-outs/ins were scored for many of the STR markers in about half of the earwax samples after a 60 days storage.

The paper reports on an unusual biological source which is very rarely considered in criminal investigations for DNA analysis. As the authors claim, it is however possible that cotton buds or, more frequently, earphones could be collected in a crime scene investigation. For this reason, it is worth knowing if autosomal STR profiles useful for individual identification could be obtained from this biological source even after some time.

Specific remarks

In the Introduction and in the Discussion sections, the authors claim that most of the amplification artefacts (drop outs and drop ins) highlighted in the earwax samples stored after 60 days could be resolved by using the most appropriate DNA extraction protocol. Do the authors know if an appropriate DNA extraction protocol for earwax samples exist? Or, how can they say that other protocols could be more appropriate? In my opinion, the only way to deal with PCR stochastic artefacts is to replicate PCR amplifications in the same analytical conditions. As the authors performed single amplifications of the samples, the readers could be interested in figuring out if all the peak heights of the surviving alleles for the heterozygous genotypes fall below the stochastic threshold? This detail should be added to the text together with the rfu value of the stochastic threshold.

According to the dropins, the authors claim that the most reliable explanation for their findings is contamination from exogenous material even if it involves only few loci in some samples. In the discussion, they say that “The reason for this is unclear, but recent work on extra-cellular DNA may shed some light on the source of these fragments”. After this sentence, the reference paper is missing and I would suggest the authors to add the reference for this paper (10.1016/j.fsigen.2017.04.015) where a possible explanation for the dropin events is reported.

I could not get the meaning of the following sentence and the relevance of the corresponding reference “Generally, the differentiation of drop-in and drop-out has never been satisfactorily resolved and some software (LikeLTD) may produce a likelihood ratio favoring the prosecution hypothesis [18]”. I can easily get the difference from dropout and dropin events! Maybe, it is better to re write the sentence to better clarify the meaning. Reference [18] is a general paper on the structure and population genetics of STRs selected for human identification not dealing with dropouts and dropins.

Table 6 and 7 could be easily assembled in a single Table as they report exactly the same complete profiles. In the Table legend, it could be described that the reference profiles of the 15 Saudi volunteers are the same as the ones of the earwax samples up to 30 days.

Are the volunteers aware that their complete autosomal STR profiles will be published in a scientific paper? Has this issue been specified in the consensus form? Have they been informed that, in the present form, individual matches (with the possibility of re-identification by linking with data in other databases) or inferences about family members are possible (especially for the family members of the two brothers missing the D18S51 locus)?

In the Results, it is not clear how the 21 allelic loss were calculated because, in Table 8, there is a chaotic mixture of locus/allele dropouts and dropins. In addition, it should be clarified the meaning of “allelic alterations” and why this has been referred only to the five situations described in the text as I can see other similar situations identified, on the opposite, as allelic losses (for instance, what’s the difference between the allelic loss in sample 6, marker D5S818, and the allelic alteration in sample 13, for the same marker? They both lost one of the two alleles of the corresponding heterozygous genotype).

In the same Table, it is not clear why the authors identified as allelic losses the genotypes in sample 6, markers TH01 and D2S1338. In the reference sample, they show homozygous genotypes. How is it possible to say that one of the two identical alleles was dropped out? Please check the genotype of marker D21S11, sample 9 which is not highlighted despite the presence of a dropout and a dropin event.

In the Results, the authors state that, in sample 9, marker D5S818, an “allelic alteration” changed the original genotype 13-13 to 9-13 while, in Table 8, the reported genotype is 7-13.

Finally, In the Materials and Methods section, some information is missing. In particular, the significance level and the software used for the statistical analyses are not reported. In addition, p. 3, lines 116-118, Shapiro-Wilk test does not give information about the different DNA concentration among samples; to this aim, the authors could report the test described in the Results.

In the Results, Table 2, the median value for the “15-days sample” is higher than the maximum.

In the Results, Table 3 could be removed. This information could be reported in the text.

P. 4, line 149, the post hoc test used should be reported.

Reviewer 2 Report

The authors present an interesting manuscript dealing with the analysis STR profiling of earwax obtained from volunteers.

The topic is of interest for the forensic genetics perspective, and it might be helpful in moving it forward.
The manuscript is well organized, the writing is clear and concise, and correct in the methodology of the research. I consider this article worthy of publication, also if and I have an issue regarding the concrete applications in forensic genetics which I hope will be useful to the Authors. 
In fact, it is a bit too vague to state in the introduction ( lines 30- 33) and in the discussion  (line 220 ) discussion "that typing DNA from earwax that could identify the suspect".

It is true that there are no publications in the literature because it is difficult to think that a suspect of a crime leaves a cotton bud with which he cleaned his ears at the crime scene.

I'm more inclined to believe that they can be used, for example, earphones.

For this reason it would be important in the Introduction, that the Authors identify some case scenarios to better clarify the usefulness of analyzing this biological material with forensic purpose.

The Authors analyze also saliva swab of the volunteer as reported in table 6, but there is not mention of the procedure in materials and methods. I think that they should added some indication.

Round 2

Reviewer 1 Report

Most of the points highlighted by this reviewer have been acknowledged but few still need to be clarified.

POINT 1 –

a) the question still stands about what the authors claim; they say that using an appropriate DNA extraction method, most of the amplification artefacts can be overcome. Do the authors know if an appropriate DNA extraction protocols for earwax samples exist? If not, I would suggest to delete this sentence or to clarify the meaning of the sentence.

b) In addition, I don’t agree completely with the definition given by the authors in the Introduction paragraph of the revised version about the analytical and stochastic thresholds.

The analytical and stochastic threshold values are defined following internal laboratory validation on each multiplex-PCR system and typing procedure used in the laboratory (see, for example, SWGDAM Interpretation Guidelines for Autosomal STR Typing by Forensic DNA Testing Laboratories). For this reason, it’s meaningless to say that they are “typically” 50 and 200 rfu, respectively. Every lab sets its own values, determined according to internal validation procedures. Have the authors performed these validation procedures on the PCR multiplex system selected for their study? If yes, they have to report the corresponding values.

For the above mentioned reason, in the revised version I’m suggesting to delete the following sentences: “which is typically set to 50 rfu” and “with approximate 200 rfu”. Similarly, in the discussion paragraph, I suggest to delete the mention to “rfu 200 [11]”.

c - I can’t get the meaning of the following sentence added to the discussion paragraph of the revised version: “In discussion, we added: in the present study, the peak heights of the surviving alleles of the heterozygous genotypes fall above the stochastic threshold, rfu 200 [11] and their sister alleles approaches the analytical threshold 50rfu [9].

If the peak height of the surviving allele of the heterozygous genotype fall above the stochastic threshold (ST) and the corresponding sister allele approaches the analytical threshold, this is not an allele dropout but it is an allelic imbalance as both alleles are visible.

d- Similarly, I can’t get the meaning of the following sentence: “Drop out was likely in the heterozygous genotypes approaching the stochastic threshold, while their sister alleles were below the analytical threshold”. Please clarify and rewrite the sentence.

e) The original question for the authors was the following: the peak heights of the alleles recorded in samples 4, 6, 8, 9, 10 and 13 after 60 days and highlighted in orange fall above or below the stochastic threshold (ST)? In the case that all of them fall below the ST, the threshold itself has been set properly, as this is the condition for which there is the possibility that droput events have occurred (what actually happened for those markers). If one or more alleles fall above the ST, the threshold itself needs to be set more properly. After checking the corresponding reference profiles, you are now confident that your ST is set properly so that you can easily identify the homozygous genotypes below the ST as likely allelic dropouts?

POINT 3 : again, I can’t get the meaning of the following sentence: “Generally, the differentiation of drop-in and drop-out has never been satisfactorily resolved and a probabilistic approach may produce a likelihood ratio favoring the prosecution hypothesis [22]”. The cited paper illustrates the use of a probabilistic approach incorporating the probability of allelic drop-out to deal with low template DNA samples and I can’t get the connection with the starting sentence.

POINT 7: I wrongfully indicated sample 6 instead of sample 8 but the question still stands: how is it possible to highlight as allelic losses the genotypes for markers TH01 and D2S1338? In the reference profiles the genotypes are homozygous. How is it possible to say that one of the two identical alleles is dropped out?
